# Anti-Inflammatory and Pro-Regenerative Effects of Hyaluronan-Chitlac Mixture in Human Dermal Fibroblasts: A Skin Ageing Perspective

**DOI:** 10.3390/polym14091817

**Published:** 2022-04-29

**Authors:** Alice Donato, Elisa Belluzzi, Elena Mattiuzzo, Rina Venerando, Massimiliano Cadamuro, Pietro Ruggieri, Vincenzo Vindigni, Paola Brun

**Affiliations:** 1Histology Unit, Department of Molecular Medicine, University of Padova, 35121 Padova, Italy; alice.donato@unipd.it (A.D.); elena.mattiuzzo85@gmail.com (E.M.); 2Musculoskeletal Pathology and Oncology Laboratory, Department of Surgery, Oncology and Gastroenterology (DiSCOG), University of Padova, 35128 Padova, Italy; elisa.belluzzi@unipd.it; 3Orthopaedics and Orthopaedic Oncology, Department of Surgery, Oncology and Gastroenterology (DiSCOG), University of Padova, 35128 Padova, Italy; pietro.ruggieri@unipd.it; 4Department of Molecular Medicine, University of Padova, 35121 Padova, Italy; rina.venerando@unipd.it (R.V.); massimiliano.cadamuro@unipd.it (M.C.); 5Department of Neuroscience, University of Padova, 35100 Padova, Italy; vincenzo.vindigni@unipd.it

**Keywords:** hyaluronan, chitlac, dermis, skin ageing, reactive oxygen species, inflammation, extracellular matrix

## Abstract

Inflammation and the accumulation of reactive oxygen species (ROS) play an important role in the structural and functional modifications leading to skin ageing. The reduction of inflammation, cellular oxidation and dermal extracellular matrix (ECM) alterations may prevent the ageing process. The aim of this study is to investigate the expression of pro-inflammatory markers and ECM molecules in human dermal fibroblasts derived from young and middle-aged women and the effects of lactose-modified chitosan (Chitlac^®^, CTL), alone or in combination with mid-MW hyaluronan (HA), using an in vitro model of inflammation. To assess the response of macrophage-induced inflamed dermal fibroblasts to HA and CTL, changes in cell viability, pro-inflammatory mediators, MMPs and ECM molecules expression and intracellular ROS generation are analysed at gene and protein levels. The expression of pro-inflammatory markers, galectins, MMP-3 and ECM molecules is age-related. CTL, HA and their combination counteracted the oxidative damage, stimulating the expression of ECM molecules, and, when added to inflamed cells, restored the baseline levels of IL-1β, TNF-α, GAL-1, GAL-3 and MMP-3. In conclusion, HA and CTL mixture attenuated the macrophage-induced inflammation, inhibited the MMP-3 expression, exhibited the anti-oxidative effects and exerted a pro-regenerative effect on ECM.

## 1. Introduction

Skin ageing is a slow natural process involving deep modifications in skin strength, elasticity and hydration, mainly due to changes in the dermal compartment that lead to decreased synthesis or degradation of extracellular matrix (ECM) molecules [1]. Fibroblasts, the most abundant cellular component of the dermis, are responsible for the synthesis and maintenance of ECM composition [2]. With increasing age, intrinsic and extrinsic factors, such as ultraviolet (UV) radiation, air pollution and the human microbiome, result in the accumulation of senescent cells and, consequently, structural alterations to dermal ECM composition [3,4]. The accumulation of senescent cells also leads to the production of pro-inflammatory cytokines and reactive oxygen species (ROS) that modify the micro-environment of the skin, leading to a chronic inflammatory state (inflammaging) and contributing to ECM disruption [5,6]. Indeed, these deep dermis changes result in the increased production of matrix-degrading enzymes and decreased ECM components such as collagen, elastin, hyaluronan (HA) and glycoproteins [7,8]. Recent studies have shown that macrophage-mediated chronic skin inflammation may be considered one of the main drivers of skin ageing, causing the release of pro-inflammatory mediators and ROS that are implicated in ECM degradation [9,10]. The fibroblasts, in turn, start to produce cytokines, such as Interleukin-1β (IL-1β), tumor necrosis factor α (TNF-α) and catabolic enzymes such as metalloproteinase 3 (MMP-3), which accelerate the degradation of ECM components, principally collagen type I, collagen type III and elastin [11,12]. Furthermore, recent in vitro and in vivo studies also showed that galectins, pro-inflammatory molecules that bind specifically to β-galactoside sugars, are over-expressed during cellular senescence [13,14]. This recent knowledge of the mechanisms that lead to skin ageing indicates that therapeutic strategies that combine the reduction of inflammation and the prevention of ECM degradation may restore correct skin homeostasis. HA is a long, unbranched polysaccharide composed of repeating D-glucuronic and *n*-acetyl-D-glucosamine disaccharides. It is widely used pharmacologically in ophthalmology, rheumatology and dermatology [15]. It is well-known for its anti-inflammatory and antioxidant properties and its capacity to affect ECM production [15,16]. Chitlac^®^ (CTL) is a lactose-modified chitosan that is able to interact with both HA (Figure 1) and Galectin-1 (GAL-1). The interactions between HA and CTL were recently studied, and it was demonstrated that the complex showed a close interaction and high viscosity that could be degraded by specific enzymes, such as hyaluronidase and lysozyme [17]. Using the combination of HA and CTL to prolong and improve the therapeutic activity of HA was recently shown to attenuate macrophage-induced inflammation, MMP expression and ROS production [17,18,19,20]. Based on this finding, we hypothesized that the mixture of HA and CTL could reduce chronic skin inflammation and, consequently, prevent skin ageing. 

The aims of this study were (1) to assess and compare the expression of pro-inflammatory markers, MMPs and ECM molecules in primary human dermal fibroblasts derived from a young and middle-aged woman donor; (2) to evaluate ECM molecule production by primary human dermal fibroblasts stimulated with HA and CTL; finally, (3) to assess and compare the production of ROS, the expression of pro-inflammatory molecules, MMPs and ECM molecules in primary human dermal fibroblasts incubated with a macrophage-derived inflammatory stimulus in the presence/absence of HA and CTL.

## 2. Materials and Methods

### 2.1. Drugs, Chemicals and Cells

Medium molecular weight HA (800–1000 kDa, HTL, Javene, France) and Chitlac ^®^ (CTL, Jointherapeutics, Como, Italy) were dissolved in phosphate saline and pH was adjusted to 7.4. 

Human dermal fibroblasts isolated from the abdomens of two healthy women, a 30 (SK0355) and a 54 (SK0235) years old, were purchased from CTI Biotech (Lyon, France) and cultured using the CTIGM medium for fibroblasts (Biotech, Lyon, France), according to supplier’s instructions. The human monocyte U937 cell line was purchased from Thermo Scientific (Wilmington, DE, USA) and cultivated in RPMI (Euroclone, Pero, Italy) with 10% fetal bovine serum (FBS) (Gibco, ThermoFisher) and 1% antibiotics. Cells were then cultured at 37 °C with 5% CO_2_.

### 2.2. Activated U937 Monocyte Conditioned Medium

Human monocytes U937 were treated with 1 µg/mL lipopolysaccharide (LPS, Sigma) for 1 h after being differentiated into macrophages with phorbol 12-myristate 13-acetate (PMA) (Sigma-Aldrich) at a final concentration of 50 ng/mL for 48 h. Then, cells were rinsed and cultured for 24 h in complete RPMI to produce inflammatory conditioned medium (CM), which was collected, centrifuged, filtered and utilised to treat primary human cell cultures as previously reported [20,21]. The transition of monocytes to macrophages was analysed by inverted phase-contrast microscope observation and mRNA expression of the macrophage differentiation marker CD68 (Appendix A).

### 2.3. Evaluation of the Effects of HA and Chitlac (CTL) on Human Dermal Fibroblasts Viability

The effects of HA and CTL, alone or in combination were assessed on primary human dermal fibroblast viability. Cells were seeded at a concentrations of 7000/cm^2^ in 96 well plates and cell viability was assessed using the MTT assay (3-4,5-dimethylthiazol-2-yl-2,5-diphenyltetrazolium bromide, Sigma, MO, USA) after 24 h, 3 days and 6 days, according to a modified Denizot method [22]. The absorbance values were determined using Infinite F200 (TECAN). Each experiment was performed three times in triplicate. After this preliminary in vitro viability study, the HA and CTL concentrations for the subsequent cell culture treatments were chosen.

### 2.4. Analysis of the Anti-Inflammatory, Antioxidant and Regenerative Effects Induced by HA and CTL

The effects of HA, CTL and their mixture were analysed on fibroblasts obtained from the dermis of young (SK0355) and middle-aged (SK0235) women, incubated or not with the CM of activated U937, by detection of reactive oxygen species (ROS) production and by analysis of pro-inflammatory and ECM molecular expression. 

#### 2.4.1. Intracellular ROS Generation

Cultures of human dermal fibroblasts SK0355 and SK0235 were exposed to the CM of activated U937 cells and then treated with HA, CTL or their mixture. At 4 h after incubation, fibroblasts were harvested and the production of intracellular ROS fluorometrically determined using 2′,7′-dichlorodihydrofluorescein diacetate (H2DCFDA; Molecular Probes, ThermoFisher, CA, USA), a nonfluorescent probe that is rapidly oxidised to fluorescent 2′,7′-dichlorofluorescein in the presence of ROS.

After exposure to 5 µM H2DCFDA in warm PBS for 30 min at 37 °C, cells were washed and placed in PBS. Each experiment was performed in quadruplicate. Excitation and emission wavelengths of 488–530 nm, respectively, were used to assess fluorescence in a BD LSRFortessa X-20 Flow Cytometer (Becton, Dickinson and Company, Franklin Lakes, NJ, USA) [20].

#### 2.4.2. RNA Isolation and qPCR Analysis

Differences in mRNA expression of *IL-1β, TNF-α, GAL-1, GAL-3, MMP-3, collagen type I, collagen type III* and *elastin* genes were analysed by qPCR in primary human fibroblasts (SK0355 and SK0235) exposed or not to activated U937 CM treated with HA, CTL and their mixture for 4, 10 and 24 h.

Total RNA was extracted using TRIzol (Life Technologies, Carlsbad, CA, USA) according to the manufacturer’s instructions. RNA quality was monitored using Nanodrop 2000c spectrophotometer (Thermo Scientific) by measuring absorbance at 260/280 nm. Total RNA was incubated with DNAse I (ThermoFisher) for 15 min to remove DNA from the samples. In total, 500 ng of total RNA was reversely transcribed using Oligo-dT and Superscript II (Life Technologies, Carlsbad, CA, USA) and qPCR was carried out using Xpert quick SYBR Green (GRISP, Portugal) on a Rotor-Gene RG -3000A (QIAGEN, Germany) [20,21]. Primers used for q-PCR analysis are listed in Table 1. Gene expression was assessed using the 2ΔCt method, in which ΔCt = Ct peptidylprolyl isomerase A (PPIA)-Ct target gene.

#### 2.4.3. Enzyme-Linked Immunosorbent Assay (ELISA)

Supernatants from human dermal fibroblast cultures (SK0355 and SK0235) stimulated or not with activated U937 CM and subsequently treated with HA, CTL or their mixture were collected. IL-1β, TNF-α and MMP-3 were quantified using the ELISA kit from R&D Systems (Minneapolis, MN, USA), GAL-1 with the ELISA Kit from Ray Biotech (Ray Biotech, Inc. Norcross, GA, USA) and GAL-3 by the human ELISA kit from Boster Bio (Pleasanton, TX, USA) according to the suppliers’ protocols. Each experiment was performed in triplicate.

### 2.5. Statistical Analysis

GraphPad Prism 7 was used to conduct statistical analyses, which included an unpaired Student’s *t*-test and one-way analysis of variance (ANOVA) with Tukey’s post hoc test for multiple comparisons (San Diego, CA, USA). A *p* < 0.05 was considered statistically significant.

## 3. Results 

### 3.1. Gene Expressions of Pro-Inflammatory Molecules, Galectins, MMP-3 and ECM Molecules Are Age-Related in Cultured Dermal Fibroblasts

In order to clarify whether the expression of pro-inflammatory and ECM molecules by dermal fibroblasts is age-related, dermal fibroblasts were obtained from two healthy women of the following different ages: SK0355, a 30-year-old and SK0235, a 54-year-old. *IL-1β, TNF-α, GAL-1, GAL-3, MMP-3, collagen type I* and *III* and *elastin* expression in both cell cultures were analysed. The results showed that SK0235 cells significantly up-regulated *MMP-3*, inflammatory molecules and galectins at gene level (Figure 2a, *p* < 0.05 for *IL-1β, GAL-1*, *GAL-3*; *p* < 0.01 for *MMP-3* and *p* < 0.0001 for *TNF-α*) and at the protein level (Figure 2b, *p* < 0.05 for GAL-3 and MMP-3; *p* < 0.01 for IL-1β, TNF-α and GAL-1), when compared to the SK0355 donor. 

Furthermore, SK0235 fibroblasts from the middle-aged donor down-regulated ECM molecules when compared to SK0355 fibroblasts from the younger woman (Figure 3, *p* < 0.05 for collagen type III and elastin; *p* < 0.01 for collagen type I).

### 3.2. Viability of Fibroblasts Is Not Affected by Low Doses of HA, CTL and Their Mixture

To determine whether HA and CTL, either alone or in combination, affected the viability of human primary dermal fibroblast isolated from women of different ages, SK0355 (30 years) and SK0235 (54 years) cells were separately cultured in the presence or absence of different concentrations of the molecules (ranging from 0.25 to 1.25 mg/mL) and results are reported in Figure 3. HA, at all the tested concentrations, did not affect cell proliferation after 1, 3 and 6 days of treatment (Figure 4a,d), whereas a significant reduction of viability was found in the presence of CTL at concentrations above 0.75 mg/mL after 6 days (Figure 4b,e). Moreover, when CTL was administered to both cell cultures in the presence of 1.25 mg/mL HA (Figure 3f and Figure 4c), the viability of cells was comparable to control levels between 0.5 and 0.75 mg/mL CTL. In accordance with these findings, subsequent experiments were performed with the mixture of HA 1.25 mg/mL and CTL 0.75 mg/mL.

### 3.3. HA and CTL Upregulate the Synthesis of Collagens and Elastin 

To evaluate the effect of HA, CTL and their mixture on dermal ECM molecule expression by primary human dermal fibroblast, cells isolated from the skin of young (SK0355) and middle-aged (SK0235) women were cultured in the presence or absence of the HA and CTL and their mixture for 4, 10 and 24 h. As reported in Figure 5, the gene expression of collagen type I, collagen type III and elastin increased significantly (*p* < 0.05) in the presence of all the molecules 4 h after treatment.

### 3.4. HA-CTL Mixture Induced a Reduction of ROS Formation in Human Dermal Fibroblasts Exposed to CM of Activated U937 Cells

The antioxidant effect of the complex HA-CTL was tested on human primary dermal fibroblast cultures treated for 24 h with the conditioned medium (CM) of activated U937 cells. When cells from the skin of a young (SK0355) and a middle-aged (SK0235) woman were exposed to the CM for 24 h, a significant generation of intracellular ROS was induced when compared to that of untreated controls in both cell cultures (Figure 6, *p* < 0.001 and *p* < 0.0001). However, the subsequent treatment of SK0355 and SK0235 cultures with CTL alone or in combination with HA exerts a significant anti-oxidative effect, which was higher for CTL administered alone to cells only for SK0355 (*p* < 0.01 vs. *p* < 0.05).

### 3.5. HA, CTL and Their Mixture Exert Anti-Inflammatory Effects on Human Dermal Fibroblasts

The production of pro-inflammatory cytokines by human primary fibroblasts treated with the CM of PMA/LPS-activated U937 cells was confirmed by a significant up-regulation of IL-1β, TNF-α, GAL-1, GAL-3 and MMP-3 (Appendix A). Interestingly, we found that fibroblasts derived from SK0235, the middle-aged woman, produced higher levels of pro-inflammatory cytokines when compared to cells isolated from the younger woman (SK0355).

The anti-inflammatory effect of HA, CTL and their mixture was subsequently tested on both SK0355 and SK0235 cultures treated with the CM of the activated U937 cells for 24 h. The results showed that both molecules and their mixture induced a significant down-regulation of *IL-1β, TNF-α, GAL-1* and *MMP-3* gene expression at 4 h and *GAL-3* at 4 and 10 h after treatment in both cell cultures (Figure 7). Surprisingly, the reduction of *GAL-1* and *GAL-3* was particularly marked (*p* < 0.0001) in the presence of CTL and of the mixture and was significantly lower than that induced by HA (*p* < 0.05) in SK0235 and SK0355 cells.

The ELISA assays (Figure 8) confirmed that the CM of activated U937 human monocytes caused an increased protein expression of IL-1β, TNF-α, GAL-1, GAL-3 and MMP-3 in both SK0355 and SK0235 cultures at 4–10 h after exposure. The supplementation of HA, CTL or their mixture to the cell culture medium induced a significant down-regulation of pro-inflammatory cytokine expression at the same timepoint after treatment (*p* < 0.05). Of note, at 4 h after treatment in the cell cultures of the middle-aged donor (SK0235), there was an improved reduction of IL-1β expression with the mixture of HA-CTL when compared to the reduction induced by HA or CTL alone (*p* < 0.001 and *p* < 0.05, respectively).

## 4. Discussion

Skin ageing is characterised by several structural and functional modifications of dermal ECM and by low-grade chronic inflammation, “inflammaging” [23,24]. Recent studies have reported that macrophages are the main immune cells present in the dermis during skin ageing and that pro-inflammatory mediators may contribute to altering the ECM composition, with a progressive reduction of type I and type III collagens and elastin fibres. Among them, IL-1β is necessary for tissue homeostasis and protection, particularly in collagen production [25] and galectins, which were recently associated with inflammation and tissue degeneration [26,27]. In particular, the increased expression of MMP-3 has been shown to enhance the degradation of elastic and collagen fibres responsible for the tensile strength and elasticity of the skin, mainly of collagen type I and collagen type III [28,29]. Our previous results demonstrated that HA and CTL are able not only to neutralise the up-regulation of pro-inflammatory mediators and galectins, but also to strongly decrease ROS production and MMP-3 gene expression and restore ECM molecule production [18].

In this study, we found that dermal fibroblasts’ expression of pro-inflammatory and ECM molecules is age-related. Indeed, gene and protein up-regulation of IL-1β, TNF-α and MMP-3 was found in cells isolated from a middle-aged woman of 54 years compared to the levels observed in cells isolated from a younger woman. This age-related increase in pro-inflammatory mediators is consistent with previous studies that demonstrated a chronic low-grade inflammation during physiological skin ageing that is also recognised as a pathogenic factor in the development of several age-associated diseases [30,31,32]. Furthermore, our results showed that GAL-1 and GAL-3 were over-expressed in aged skin. To the best of our knowledge, the production of galectins has never been tested in dermal fibroblasts of women of different ages. However, many studies reported that they are associated with inflammation, tissue degeneration and up-regulation of several enzymes involved in matrix degradation in cartilage, lung, liver, pancreas and kidney parenchyma [33,34,35,36,37]. Interestingly, we found that the age-related down-regulation of *elastin and type I* and *type III collagen* expression by dermal fibroblast cultures was associated with the overexpression of pro-inflammatory molecules and galectins. Further studies should focus on the upstream mechanisms that lead to increased production of galectins in dermal inflammation. However, our study demonstrated that HA and CTL, alone or in combination, stimulate ECM production, likely contributing to the skin’s biomechanical properties (i.e., elasticity) and, importantly, this effect is retained in young and middle-aged fibroblasts. In particular, these substances induced increased mRNA expression of *collagen type I, collagen type III* and *elastin* in both young and middle-aged fibroblasts. Since monocytes/macrophages are key players in the skin ageing process [38,39], the present study also investigated the benefits of HA, CTL and their mixture in reducing the inflammatory effects of the macrophage-derived conditioned medium used to stimulate fibroblast cell cultures obtained from the dermis of women of different ages. After assessing human dermal fibroblasts’ viability in the presence of different HA and CTL concentrations, 1.25 mg/mL HA and 0.75 mg/mL CTL were selected to evaluate the response to macrophage-derived inflammatory mediators. We found that the inflammatory macrophage stimulus induced an increase in both pro-inflammatory molecule production (IL-1β, TNF-α) and galectins (GAL-1 and GAL-3) and that this response was higher in human dermal fibroblasts of middle-aged women compared to younger women, as if ageing could predispose human dermal fibroblasts to be more responsive to inflammation. We also observed the tendency towards greater production of ROS by fibroblast cultures derived from the skin of the older women, but this increase was not significant. Further experiments should be performed to better unravel this issue.

We found that the mixture of HA-CTL reduced the expression of IL-1β in comparison to that induced by HA or CTL alone in the cell cultures of middle-aged donors. Indeed, it is well-known that ageing cells up-regulate the expression of cytokines such as IL-6 and IL-1β and MMPs [11,40], and functional differences between dermal fibroblasts isolated from young and middle-aged donors were recently demonstrated in in vitro studies. In particular, up-regulation of MMPs and an increase in oxidant species were found in fibroblasts isolated from elderly donors [41]. Moreover, Wolf et al. [42] showed that human skin fibroblasts from elderly donors stimulated with LSP or cytomegalovirus infection produced higher amounts of IL-6 as well as IL-8 than fibroblasts from younger donors, supporting our finding that the level of protein secretion is dependent on the chronological age of the fibroblasts. All these findings show that the mixture of HA-CTL, combining its anti-inflammatory and protective effects with the capability of restoring ECM molecule levels, could have the therapeutic potential to improve dermis repair as well as likely counteract inflammaging.

Moreover, as previously reported for articular tissues, the mixture of HA and CTL exerted both antioxidant and anti-inflammatory effects, reducing gene and protein expression of IL-1β, TNF-α, GAL-1 and GAL-3 [19,20]. In particular, we found that CTL and the mixture HA-CTL are able to strongly reduce GAL-1 and GAL-3 expression in comparison to HA alone. It is conceivable that its effect might be linked to the interaction of the galectins with the lactose residues of CTL, which in turn inhibits the binding to the cell surface. However, additional studies are needed to better investigate the mechanisms that underload the reduced production of galectins by HA and even more CTL and HA-CTL mixtures. The reduced inflammation is also accompanied by a decrease in MMP-3, which further underlines the synergistic actions of HA and CTL in our in vitro pro-inflammatory model. Furthermore, we demonstrated that the oxidative damage was counteracted by CTL and HA-CTL administrated to cell cultures at the moment of in vitro induced oxidation. Taken together, these results confirmed our previous findings about the ability of CTL and HA to reduce both inflammation and oxidative stress in ROS-injured chondrocyte cultures [20]. The correlation between inflammation and oxidative damage in inducing cellular senescence and ECM molecule degradation, such as that of collagen and elastin, is well known [43,44]. In skin ageing, fibroblasts are mainly responsible for ROS generation, which leads to an imbalance between ECM molecule synthesis and degradation, and inflammatory mediators can be reduced by antioxidants [45]. Thus, the treatment of dermal fibroblasts with HA-CTL has the potential to reverse in vitro-induced oxidation and inflammation and promote ECM remodelling.

The main limitation of this study is that differences among multiple individuals are lacking since experiments were performed on primary fibroblasts isolated from one young and one middle-aged woman. Additional studies should be considered to analyze differences among primary fibroblasts isolated from different donors. 

## 5. Conclusions

In conclusion, this study supports the potential benefits of the HA-CTL treatment for ageing skin and could contribute to improving our knowledge in the development of new treatments toward the prevention/reduction of inflammation, oxidative stress and skin ECM maintenance and the repair. Inflammatory cytokines (*IL-1*β, *TNF*-α), *GAL-1*, *GAL-3* and *MMP-3* levels increased in middle-aged fibroblasts compared to young fibroblasts supporting the fact that there is an increase in inflammation during ageing. The production of ECM molecules (*COL-I, COL-III* and *ELN*) decreased in the middle-aged fibroblasts compared to young ones. The middle-aged fibroblasts respond more strongly to an inflammatory stimulus than young fibroblasts, producing higher levels of inflammatory cytokines but comparable levels of ROS. The HA and CTL treatment mitigated the macrophage-induced inflammation exhibiting anti-oxidative and anti-inflammatory effects. Indeed, a decrease in the levels of inflammatory cytokines, galectins, MMP-3 and ROS was observed in young and middle-aged fibroblasts.

Further in vitro and in vivo investigations should be performed to further understand the mechanisms of action of these molecules on human skin of different donors and to support their dermatological efficacy.

## Figures and Tables

**Figure 1 polymers-14-01817-f001:**
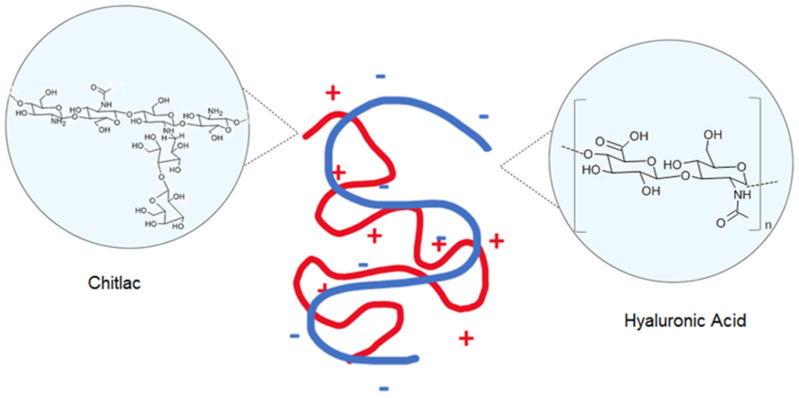
A schematic representation of the interaction between Hyaluronan (HA) and Chitlac (CTL).

**Figure 2 polymers-14-01817-f002:**
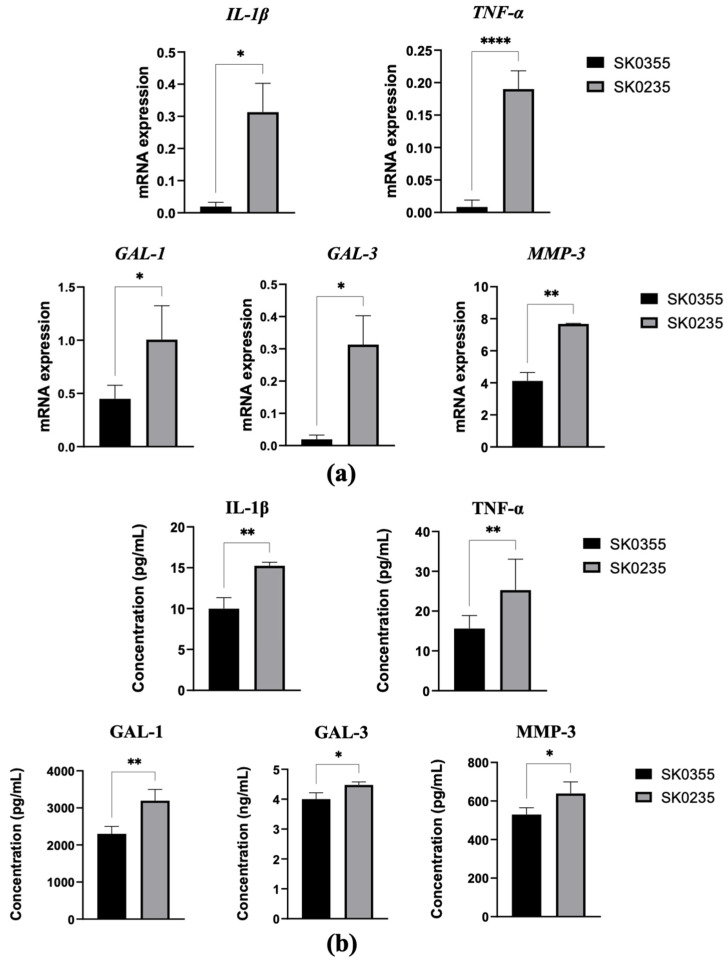
Dermal fibroblast expression of pro-inflammatory molecule, galectins and MMP-3. Dermal fibroblast expression of pro-inflammatory molecule, galectins and MMP-3. Human dermal fibroblasts SK0355 (from a 30-year-old woman) and SK0235 (from a 54-year-old woman) were cultured and the expressions of pro-inflammatory molecule, galectins and MMP-3 were analysed. RNA transcript levels (**a**) and protein expression (**b**) were evaluated by qPCR and ELISA test, respectively. Data are reported as mean ± standard error (SE) of three independent experiments. Statistical differences based on unpaired Student’s *t*-test. * *p* < 0.05 and ** *p* < 0.01 and **** *p* < 0.0001.

**Figure 3 polymers-14-01817-f003:**
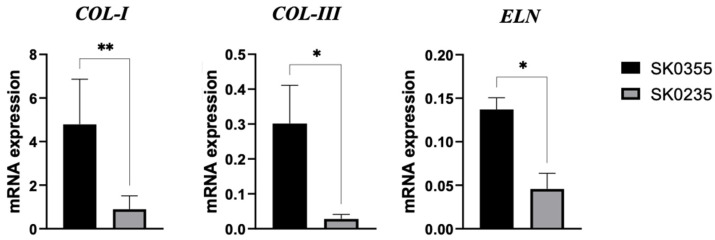
Dermal fibroblast expression of ECM molecules. Human dermal fibroblasts SK0355 from a 30-year-old woman) and SK0235 (from a 54-year-old woman) were cultured and their expression of collagen type I (COL-I), collagen type III (COL-III) and elastin (ELN) were compared. RNA transcript levels were evaluated by qPCR. Data are reported as mean ± SE of three independent experiments. Statistical differences based on unpaired Student’s *t*-test. * *p* < 0.05 and ** *p* < 0.01.

**Figure 4 polymers-14-01817-f004:**
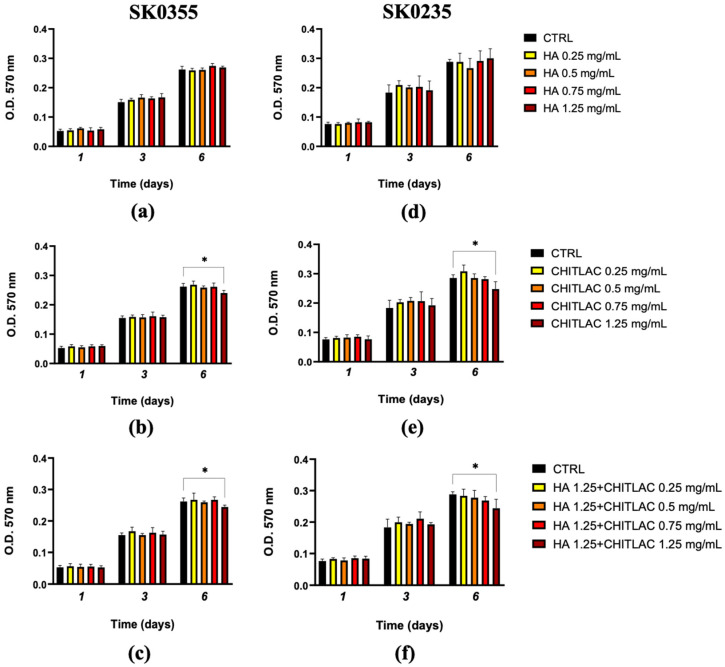
Time-dependent effects of HA, CTL and their mixture on human dermal fibroblast viability. In total, 7000/cm^2^ SK0355 (from a 30-year-old woman) and SK0235 (from a 54-year-old woman) cells were seeded in 96-well culture plates and exposed to the stated concentrations of HA (**a**,**d**), CTL (**b**,**e**), or different combinations of HA-CTL mixture (**c**,**f**) for various exposure durations. The MTT test was conducted 1-, 3- and 6-days following treatment. The results are presented as mean ± SE of three independent experiments. The unpaired Student’s *t*-test was used to determine statistical differences. * *p* < 0.05 when compared to untreated cells.

**Figure 5 polymers-14-01817-f005:**
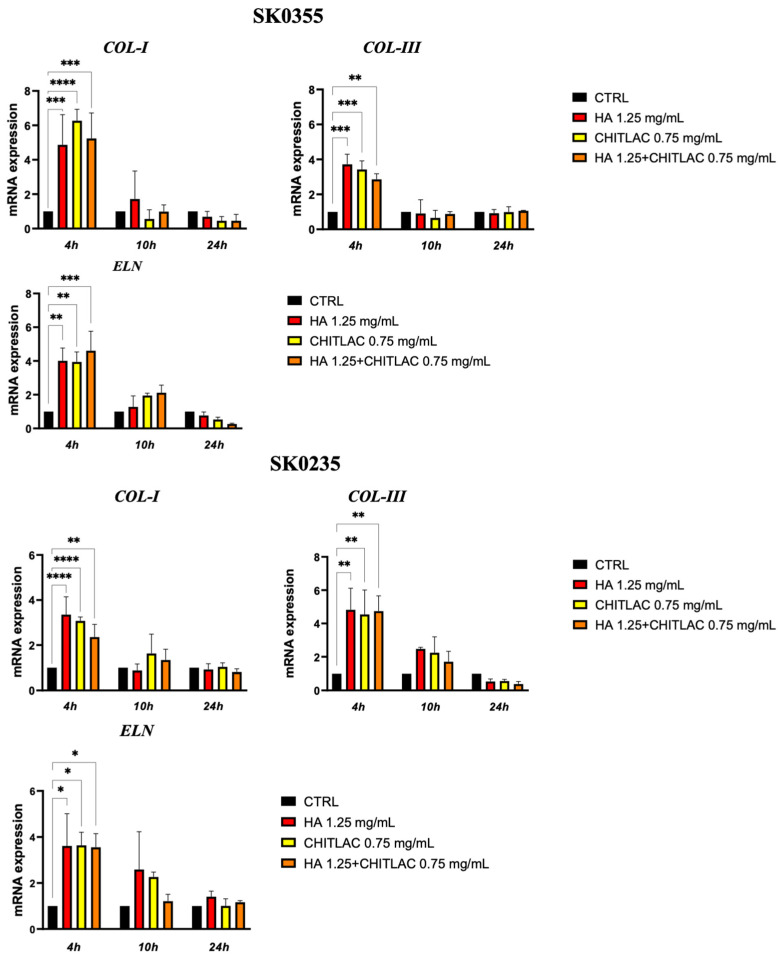
Dermal fibroblast gene expression of ECM molecules in the presence or absence of HA, CTL and HA-CTL mixture. Human dermal fibroblasts SK0355 (from a 30-year-old woman) and SK0235 (from a 54-year-old woman) were seeded in 6-well culture dishes and cultured in the presence or absence of 1.25 mg/mL HA, 0.75 mg/mL CTL or their mixture. RNA transcript levels specific for COL-I, COL-III and ELN were evaluated by qPCR as described in the Materials and Methods section at 4, 10 and 24 h after treatment. Statistical differences are based on one-way ANOVA test with multiple comparisons. Data are expressed as mean ± SE obtained from three independent experiments. * *p* < 0.05, ** *p* < 0.01, *** *p* < 0.001 and **** *p* < 0.0001 vs. respective control.

**Figure 6 polymers-14-01817-f006:**
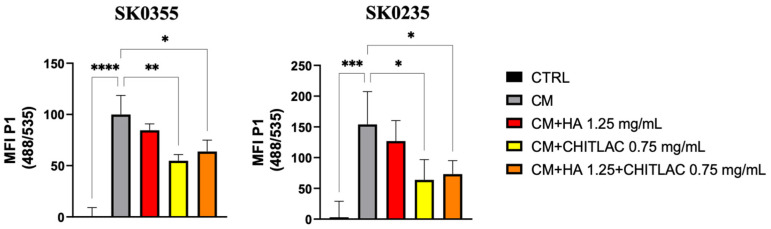
Reactive oxygen species (ROS) generation in human fibroblast cultures treated with U937 CM in the presence or absence of HA, CTL alone or in combination. Primary human fibroblasts SK0355 (from skin of a 30-year-old woman) and SK0235 (from skin of a 54-year-old woman) were exposed to U937 CM for 24 h before being treated for 4 h with 1.25 mg/mL HA, 0.75 mg/mL CTL, or a combination of the two. The intracellular ROS production of cells was then assessed using the 2′,7′-dichlorodihydrofluorescein diacetate probe and flow cytometry analysis, with at least 10,000 events collected. ROS production is measured as a percentage of the fluorescence intensity of U937 CM-treated cells. The one-way ANOVA test with multiple comparisons was used to determine statistical differences (Fisher LSD). The proportion of fluorescence seen in at least two independent trials is reported as mean ± SE, considering the control equal to 0. * *p* < 0.05, ** *p* < 0.01, *** *p* < 0.001 and **** *p* < 0.0001 vs. respective CM-treated control.

**Figure 7 polymers-14-01817-f007:**
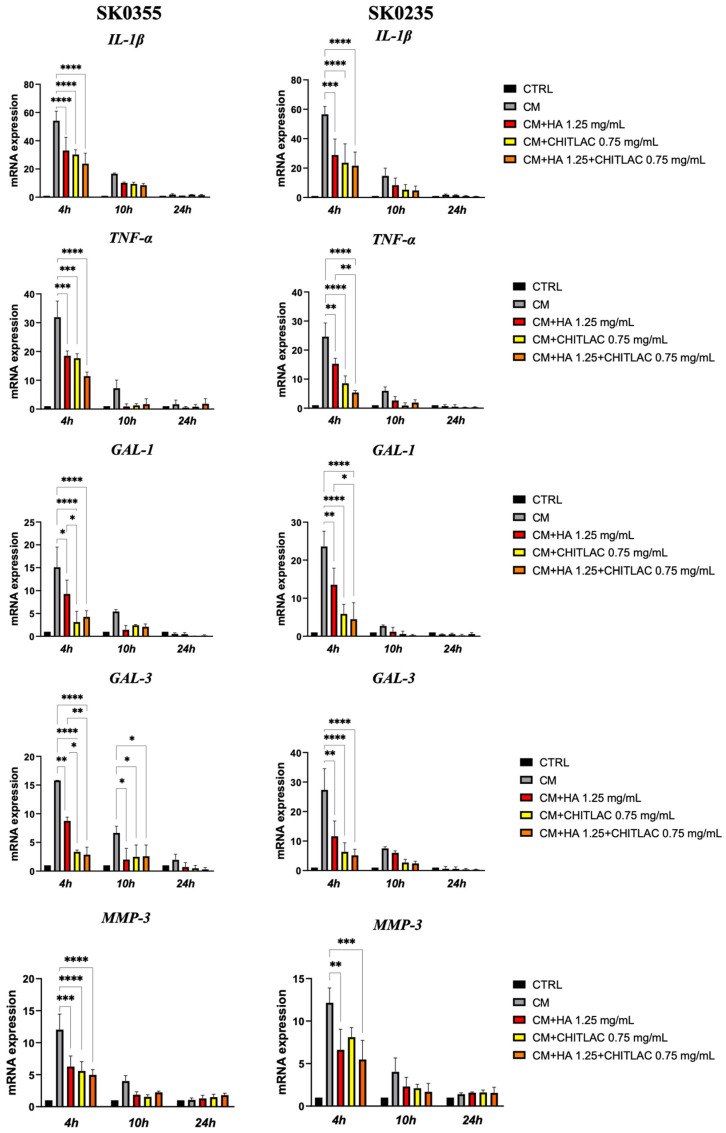
Pro-inflammatory molecule and MMP-3 gene expression in dermal fibroblasts after exposure to U937 CM in the presence or absence of HA, CTL and HA-CTL combination. Human dermal fibroblasts SK0355 (from 30-year-old woman) and SK0235 (from 54-year-old woman) were seeded in 6-well culture dishes and exposed to activated U937 CM for 24 h before being cultured in the presence or absence of 1.25 mg/mL HA, 0.75 mg/mL CTL, or a combination of the two. At 4, 10 and 24 h after treatment, RNA transcript levels specific for IL-1β, TNF-α, GAL-1 and GAL-3 were measured using qPCR as described in the Materials and Methods section. The one-way ANOVA test with multiple comparisons is used to determine statistical differences. The results are presented as mean ± SE of three independent experiments. CM stands for fibroblast cultures that have been exposed to U937 CM. CM, fibroblast cultures exposed to U937 CM. * *p* < 0.05, ** *p* < 0.01, *** *p* < 0.001 and **** *p* < 0.0001 as compared to the respective CM-treated control.

**Figure 8 polymers-14-01817-f008:**
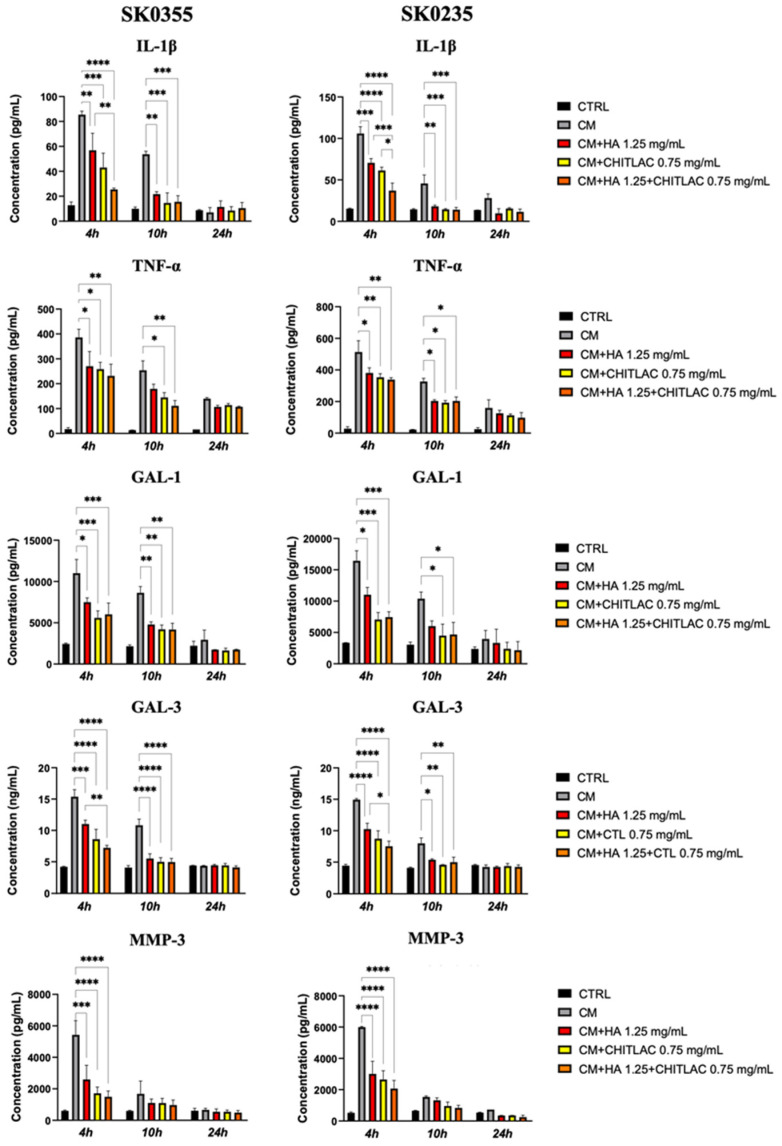
IL-1β, TNF-α, GAL-1, GAL-3 and MMP-3 protein expression in dermal fibroblasts after exposure to U937 CM in the presence or absence of HA, CTL and HA-CTL mixture. Human dermal fibroblasts SK0355 from a 30-year-old woman and SK0235 from a 54-year-old woman were cultivated in the presence or absence of 1.25 mg/mL HA, 0.75 mg/mL CTL, or their mixture after 24 h of exposure to activated U937 CM. At 4, 10 and 24 h following treatment, ELISA was used to analyse IL-1β, TNF-α, GAL-1 and GAL-3 protein expression. The one-way ANOVA test with multiple comparisons was used to determine statistical differences. The results are presented as mean ± SE of three independent experiments. CM stands for fibroblast cultures that have been exposed to U937 CM. * *p* < 0.05, ** *p* < 0.01, *** *p* < 0.001 and **** *p* < 0.0001 vs. respective CM-treated control.

**Table 1 polymers-14-01817-t001:** Genes studied and primers used.

Gene (Accession Number)	Name	Primer Sequences
*COL1A1* (NM_000088.4)	*Collagen type I alpha 1 chain*	Fw 5′-CATAAAGGGTCACCGTGGCT-3′ Rv 5′-GGGACCTTGTTCACCAGGAG- 3′
*IL-1β* (NM_000576.3)	*Interleukin 1 beta*	Fw 5′-GAATCTCCGACCACCACTACAG-3′ Rv 5′-TGATCGTACAGGTGCATCGTG-3′
*LSGALS1* (NM_002305.4)	*Galectin 1*	Fw 5′-TCTCGGGTGGAGTCTTCTGA-3′ Rv 5′-GTTCAGCACGAAGCTCTTAGC-3′
*LSGALS3* (NM_002306.4)	*Galectin 3*	Fw 5′-CTGCTGGGGCACTGATTGT-3′ Rv 5′-TGTTTGCATTGGGCTTCACC-3′
*MMP-3 * (NM_002422.5)	*Matrix metalloproteinase 3*	Fw 5′- TCACTCACAGACCTGACTCG-3′ Rv 5′-AAAGCAGGATCACAGTTGGC-3′
*ELN* (NM_000501.4)	*Elastin*	Fw 5′-CAGCTAAATACGGTGCTGCTG-3′ Rv 5′-AATCCGAAGCCAGGTCTTG-3′
*COL3A1* (NM_000090.4)	*Collagen type III alpha 1 chain*	Fw 5′-CTTCTCTCCAGCCGAGCTTC-3′ Rv 5′-CCAGTGTGTTTCGTGCAACC-3′
*PPIA* (NM_021130.5)	*Peptidylprolyl Isomerase A*	Fw 5′-GGGCTTTAGGCTGTAGGTCAA-3′ Rv 5′-AACCAAAGCTAGGGAGAGGC-3′
*TNF-α* (NM_000594.3)	*TNF- alpha*	Fw 5′-AAGCCTGTAGCCCATGTTGT-3′ Rv 5′-GGACCTGGGAGTAGATGAGGT-3′

Fw—forward; Rv—reverse.

## Data Availability

The data presented in this study are available on request from the corresponding author.

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
