# Peer review of "Anti-Inflammatory and Pro-Regenerative Effects of Hyaluronan-Chitlac Mixture in Human Dermal Fibroblasts: A Skin Ageing Perspective"

_polymers, 2022, doi:10.3390/polym14091817_

Round 1

Reviewer 1 Report

Donato et al. have investigated the anti-inflammatory and pro-regenerative effects of mixture of hyaluronan and chitlac in human dermal fibroblasts. In general, this work is methodically done and well-written with appropriate scientific interpretations. However, the following points need to be addressed before this paper could be accepted for publication:

  1. Title – it should be modified to be “Anti-inflammatory and pro-regenerative effects of hyaluronan-chitlac mixture in human dermal fibroblasts: a skin ageing perspective”
  2. Keywords – “skin” and “ageing” can be one keyword as “skin ageing”.
  3. Introduction – The aim of the study should be provided in the form of a paragraph while still using numberings like (1), (2)…
  4. It is important to provide chemical structures of hyaluronan and chitlac along with some of their basic physico-chemical characteristics, as the work is for a polymer journal.
  5. Materials and methods - each of the subsections in section 2.4 should be provided with a reference.
  6. Conclusions – some more sentences should be added to mainly emphasize the tentative/possible/proven key mechanism(s) of action of HT and/or CTL in attenuating inflammation, inhibiting MMP-3, exhibiting anti-oxidative effects and exerting pro-regenerative effect on ECM.
  7. Figures – Figures 1, 3, 4, 6 & 7 should be enlarged to make the axis labels and legends more clearly visible.
  8. References – all the references should be double-checked carefully for “Polymers” reference formatting.
  9. Funding – a grant number should be included if available.

Reviewer 2 Report

The article entitled “New directions in skin ageing: anti-inflammatory and pro-regenerative effects of Hyaluronan and Chitlac” presented by Alice Donato and co-workers represents and interesting evaluation about mid-MW Hyaluronan and lactose-modified chitosan effects on pro inflammatory mediators, MMP, collagen and elastin expression.

Although the manuscript contains so much data, it should be revised and better organized to ensure that it reaches an excellent level for its publication in Polymers magazine. Some suggestions have been reported below:

  • Abstract: The abstract is too long. The abstract is an objective representation of the article, and it should be clear and concise (200 words maximum). It must not contain so many results.
  • The authors should select most appropriate keywords (for example skin ageing could be a single keyword and inflammaging should be removed).
  • Line 23: “in vitro” should be write in italic. Please, correct and check in the entire manuscript.
  • Line 50: please, replace semicolon with comma.
  • Line 59: please, remove comma between degradation and may.
  • Line 69: the “s” after MMP should be deleted.
  • Line 72-79: Please avoid the list when reporting study objectives. Try to rephrase and bring the points back into the text, by discussing them.
  • Line 90: Please, insert full name of fetal bovine serum before acronym.
  • The authors have inserted Supplementary file. However, only the reference regarding Figure S2 was found in the main text.
  • In my opinion, too many Figures have been inserted in the main text. Figure 8, for example, could be moved in the supplementary file.
  • Please, carefully check that all references have been formatted following the journal guidelines. References should be described as follows, depending on the type of work:

For example, Journal Articles: 1. Author 1, A.B.; Author 2, C.D. Title of the article. Abbreviated Journal Name YearVolume, page range.

For example, in all references the Journal Name was not abbreviated. Please, correct and check all references.

Round 2

Reviewer 1 Report

The authors have satisfactorily addressed all the comment raised by reviewers and therefore I recommend acceptance of this article for publication in Polymers.

Reviewer 2 Report

The authors have reported all suggested changes. In my opinion the manuscript has been sufficiently improved and can now be accepted for publications in Polymers.